# Molecular Aggregation in Immune System Activation Studied by Dynamic Light Scattering

**DOI:** 10.3390/biology9060123

**Published:** 2020-06-12

**Authors:** Elena Velichko, Sergey Makarov, Elina Nepomnyashchaya, Ge Dong

**Affiliations:** 1Institute of Physics, Nanotechnology and Telecommunications, Peter the Great St.Petersburg Polytechnic University, St. Petersburg 195251, Russia; makarov@cee.spbstu.ru (S.M.); elina.nep@gmail.com (E.N.); 2School of Aerospace Engineering, Tsinghua University, Beijing 100084, China

**Keywords:** dynamic light scattering, humoral immune system, immune complexes, immune diagnostics

## Abstract

Determination of the concentration and size of the circulating immune complexes in the blood is an essential part of diagnostics of immune diseases. In this work, we suggest using the dynamic light scattering method to determine the sizes of circulating immune complexes in blood serum. By the dynamic light scattering spectrometer, we found that for healthy and sick donors, the size and concentration of circulating immune complexes differed significantly. The dynamics of formation of these complexes were also examined in this work. It was shown that the formation of immune complexes in the blood of healthy donors is faster than the same reactions in the blood serum of donors with diseases. The results can be used in the diagnostics of the immune status and detection of chronic inflammation. We can recommend the dynamic light scattering method for implementation in biomedical diagnostics.

## 1. Introduction

At present, research into the human immune system is of great interest. This system is one of the most important for homeostasis and anti-infective and anti-cancer protection of an organism [1,2]. A great number of diseases, including the most dangerous ones, such as cancer or autoimmune diseases, are closely related to the immune response functionality. Chronic immunopathological conditions affect all parts of cellular and humoral immunity, causing such pathologies [3].

To study the functionality of humoral immune response, modern medicine is studying the level of circulating immune complexes (CICs) in the blood. CICs play an important role in phagocytosis. Prolonged circulation in the blood of CICs, even with a slight increase in their level, contributes to the formation of their deposits in tissues and increased adhesion and aggregation of platelets, which in turn leads to disruption of blood circulation and blockage of blood vessels, as well as damage and tissue necrosis [4]. The duration of circulation and elimination of immune complexes depends on the functional state of the system of nonspecific resistance of an organism. The formation of circulating immune complexes occurs upon binding of albumins and immunoglobulins in the blood serum with antigens. The removal of immune complexes from the body depends on their size [5]. At high concentrations of antigens, large immune complexes are formed. As usual, they are rapidly removed in the process of phagocytosis. At low concentrations of antigen, small-sized immune complexes are formed, so they escape the attention of phagocytes. As a result, they can form deposits in the tissues, violate the microcirculation, and provoke the subsequent degradation of tissues.

Different types of analysis are applied to study human immunity. The most common method of determining the concentration of immune complexes is the ELISA test. However, it demands a set of enzymes, markers, and reagents so it is currently impossible to implement it for regular screening of the population [6,7]. The opposite disadvantages belong to hemolytic tests, used to assess the activity of the immune system in general. These tests are used as a screening analysis but do not allow revealing the specific character of the observed disruption of the immune system.

To study complex biological fluids, such as blood serum, various methods of fractionation [8] can be used. It is even possible to provide allocation of separate immune complexes, but it is extremely difficult. These techniques require a long time and/or complicated and expensive reagents. This problem has forced researchers to develop new approaches. At present, optical methods are gaining relevance in studies of humoral immunity and blood protein composition.

In several papers, it was shown that the spectral density of scattered light from the blood plasma of a healthy donor and a donor with cancer differs significantly [9,10]. More specifically, the spectral peaks in the detected scattered light signal corresponding to albumin and globulins change their relative intensity [10]. It was found that the spectral intensity of scattering allowed one to identify people at risk due to the fact that in oncology states, the relative concentrations of albumin in the blood increased and of globulin decreased. Similar studies were conducted using dynamic light scattering (DLS). A growing percentage of low molecular weight compounds (1–10 nm) and reduction of percentage of large proteins (11–30 nm) and their agglomerates in the presence of the disease were detected [11,12]. 

In [13,14,15], the Rayleigh–Debye method (static light scattering) was used for the study of molecular complexes in the blood serum. The size and molecular weight of complexes in the studied liquids were determined. It is shown that the optical properties of blood and the calculated parameters of the structures in it from healthy donors and donors with different pathologies differ. In [10,16], the methods of light scattering were used to diagnose pathological processes in the human body. They demonstrated that both concentrations of individual proteins and CICs and their size distributions change in pathology. 

The sizes of particles in solutions of biological liquids are effectively calculated with the help of the dynamic light scattering method [17,18]. This method is widely used in the study of molecular solutions and allows measuring the size of molecules, their shape, diffusion coefficients, and the degree of intermolecular interactions directly in the liquid, e.g., in blood serum, without the need for additional sample preparation [19]. The method of dynamic light scattering has been successfully applied for the analysis of sizes and diffusion coefficients of proteins [20] and protein aggregates [21]. Studies were also carried out with model solutions of blood serum [22] provided by mixing solutions of albumins and globulins in different concentrations.

The limiting factor in the application of light scattering methods in general and methods of dynamic light scattering in particular to the study the blood serum and plasma are their complex multicomponent compositions. The blood serum contains albumins, globulins, CICs, and other protein aggregates, lipoproteins, fibrinogen, a large number of minor proteins, parts of platelets, and red blood cells, as well as hundreds of low-molecular-weight compounds like amino acids, lipids, carbohydrates.

However, attempts to determine the general characteristics of the dimensional structure of serum by dynamic light scattering were described in many papers. Therefore, in [23], the possibilities of dynamic scattering as a diagnostic tool for analyzing the risks of diabetes and cardiovascular diseases are discussed. It is shown that with the considered method, it is possible to analyze concentration of protein components in the liquid. In [24], the method of dynamic light scattering was used to determine the size composition of blood serum and its changes in the process of protein degradation.

In [25], the processes of aggregation of serum proteins with the particles of composite materials used for drug delivery are demonstrated. Changes in the size of serum units with the addition of various nanoparticles have been detected. The results obtained by dynamic light scattering method are compared with those of electron and atomic force microscopy. It is shown that dynamic scattering allows correct detection of the size of proteins in the process of sample preparation.

In [26], the method of dynamic light scattering is applied for the detection of cancer markers using specially prepared probes based on gold nanoparticles. Similar studies are carried out in [27] using nanoparticles of different metals.

In previous studies, it was demonstrated that the resolution of DLS is extremely large and allows us to measure particle sizes from 0.5 nm to several microns [28]. The size of the investigated molecular structures in the blood usually lies in the range of 1–500 nm. Thus, it has been shown that the dynamic light scattering method allows one to draw conclusions about the functional properties of proteins, to determine concentrations of CICs and individual proteins, and to analyze the processes of the structure formation in biological fluids. The measurement of these parameters has practical significance for modern medicine. 

In this work, we propose applying the modified DLS method. It was developed by authors to study precisely polydisperse molecular solutions, for instance blood serum. The developed DLS method allowed us to determine the size of the molecular aggregates, including CICs and also individual protein fractions in blood serum. We also suggest applying the DLS to analyze the formation of CICs and protein aggregates while immune response. As proposed, the DLS method has no demand for special reagents; it greatly simplifies the procedure and reduces the cost of research. Therefore, it provides a great potential for the DLS method in medical diagnostics and biology for the investigation of blood serum proteins and their complexes. 

## 2. Materials and Methods

### 2.1. Dynamic Light Scattering Spectrometer

To conduct the DLS measurements of the size of molecules and molecular aggregates in biological fluids, it was proposed to use the DLS spectrometer developed by the authors (Figure 1). 

The optical system developed for this scheme was described in detail in [29] and allowed obtaining maximum signal intensity while maintaining spatial scattering coherence. For the development, we used the scattering theory described in [30]. A single-frequency semiconductor laser module with a coherence length of more than 100 m, wavelength λ = 633 nm, and radiation power of up to 10 mW was used as a radiation source [31]. The choice of the wavelength of the optical radiation source is determined from the requirement to minimize the absorption of the test solution. The absorption spectrum of blood serum was studied in [32,33]. The circuit elements selected according to calculated requirements provided signal-to-noise ratio (SNR) of more than 10 for all detected signals [28]. SNR was calculated using [34,35,36]. The choice of the photomultiplier is discussed in detail in our recent article [36] and is explained by main features of the spectrometer, which are very weak scattered radiation signals (~−70 dBm), low-frequency nature of the received signals (<100 kHz), and low background illumination due to the use of optical fibers in the optical scheme.

To process the experimental data, special mathematical algorithms should be used [37,38]. The regularization algorithms do not require the knowledge of the distribution shape and include reasonable physical requirements of non-negativity of results. Among the variety of regularization methods, the CONTIN method was most popular until recent times [39]. Unfortunately, it is very demanding for the choice of regularization parameter and does not allow the resolution of narrow peaks. In this context, we decided to use a more advanced algorithm based on the Tikhonov regularization method [40]. This algorithm was designed by the authors specifically for the study of molecules and molecular aggregates in biological fluids, which usually include narrow close-standing peaks. Knowing the diapason of molecular aggregate sizes in blood serum, we reduced the calculation interval, which allowed us to get higher resolution and reduce the time of the experiment. In addition, we reduced the number of points in each peak; usually, we used 20 points to avoid their blurring. By successive iterative solution of the problem, the number of points is reduced and the central values of particle sizes in the population are found that correspond to the maximum scattering intensity. As a result, the developed algorithm allowed analyzing sizes and concentrations of more than 4 components in fluid. In the commercial DLS spectrometers, the maximum number of components correctly resolved by the algorithm does not exceed 3. The testing of the algorithm and the used spectrometer is described in [28]. It was shown that the error in determining the size of structures in a liquid is less than 8%.

The temperature while measurements were made was set by heat stabilizer with ±0.1 °C deviation. All samples were placed in a square-cut glass cell and set in a heat stabilizer before and during measurements. The size of the cell was 1 × 1 × 2 cm, with optical pathlength inside the cell equal to 0.8 cm. To achieve the maximum signal-to-noise ratio, it was necessary to match the size of the light scattering region, determined by the waist length of the focused laser beam, with the size of the space projected onto the photodetector part of the circuit. The latter was determined by the parameters of the lens and diaphragm in front of the photomultiplier and was chosen as 6.3 mm. Minimal volume of the studied sample was 0.5 mL. A laser beam illuminated the cell, scattered radiation at the 90° angle was transferred by the system of the diaphragm (1.5 mm), step-index multimode fiber, and fiber collimator to photoelectronic multiplier Hamamatsu H11706-01. 

The photocurrent at the output of the photodetector i(t) is proportional to the intensity $I_{s}\left(t\right)$ averaged over the recorded scattering signal:$\;i\left(t\right)\sim \;\langle E_{s}\left(t\right)E_{s}^{*}\left(t\right)\rangle \;\sim \;I_{s}\left(t\right)$, $E_{s}\left(t\right)$ is the electric field value in the detection plane. The spectrum of the photocurrent i(ω) is shifted to the low-frequency region. Autocorrelation function G(τ) of such a signal is related to the photocurrent spectrum by the Fourier transform and carries information about the size of the scatterers. Normalized autocorrelation function g(τ) is written as exponential functions with different degrees, defined by diffusion coefficients *D*_*i*_, and different amplitudes, defined by concentration of particles *N*_*i*_. While analyzing the experimental data, we calculate relative concentration of scatterers of each size *N*(*d*) from set diapason *d*_min_
*d*_max_ with set resolution Δ*d*.
(1)$$i\left(\omega \right)=\frac{1}{2\pi }\int G\left(\tau \right)\exp\left(-i\omega \tau \right)d\tau ,$$
(2)$$g\left(\tau \right)=e^{-2q^{2}D\tau },$$
where q=(4πn/λ)sin(θ/2) is the scattering vector, *n* is the refractive index of the medium, λ is the illuminating light wavelength, *D* is the diffusion coefficient, and θ is the angle of scattering detection. The revealed relationship allows one to establish the diffusion coefficient and to determine the sizes of nanoparticles $d=\frac{k_{B}T}{3\pi \eta D}$, where $k_{B}$ is the Boltzmann constant, *T* is the temperature, η is the viscosity of the liquid, and *d* is the hydrodynamic diameter of the scatterers. This equation is obtained using the Stokes–Einstein equation [18]. 

The proposed scheme of DLS spectrometer allowed us to achieve high SNR of more than 10 for all detected signals; for example, the SNR value for blood serum sample in minimally used concentration was equal to 28. The spectrometer was implemented in a compact size of 25 × 15 × 5 cm with a weight less than 2 kg [29]. 

The developed algorithm based on the Tikhonov regularization method for processing of experimental data allowed us to calculate size distributions with an error not exceeding 8%. We compared the results with CONTIN algorithm results, which are used in commercial DLS spectrometers. The developed algorithm proved a significant advantage when the number of components in the studied fluid is more than 3 [28]. 

Testing of the developed hardware–software complex was carried out in the study of suspensions of glass microspheres, solutions of the protein albumin, and suspensions of metallic nanoparticles used in medical therapy and diagnostics. Supporting studies were conducted by scanning electron microscopy. The error in determining the size of nanoparticles in polydisperse fluids was less than 10% with a confidence level of 95%.

### 2.2. Experimental Samples

In this paper, the solution of blood serum albumin was used as a model of biological fluid. Albumin accounts for about 40% of all proteins of human serum and has an important role in the human body, defined by a wide variety of functions. Albumins provide a colloid osmotic pressure of blood, regulate blood pH, act as carriers for hormones, minerals, toxins, and drugs and, most importantly, take part in the form of CICs.

Human serum albumin (HSA) dissolved in distilled water in a concentration of 1 g/L was used to measure the size and formation dynamics of aggregates in albumin solution. To provoke protein aggregation and formation of protein structures, the measurements were also carried out with solutions of albumin with various pH. For instance, when adding 0.5 µL 0.1% NaOH solution in 1 mL of aqueous albumin solution, the acidity was 8. Acetic acid (CH_3_COOH) and sodium chloride (NaCl) were added to the solution to obtain lower pH values. To examine the binding properties of HSA molecules, the stabilized monodisperse suspension of 20 nm gold nanoparticles was added to neutral pH albumin solution. The experimental results are presented in Section 3.1.

Blood serum was taken from apparently healthy donors and donors with diabetes mellitus and oncology (2nd and 3rd stages of prostate cancer) who participated voluntarily. Blood samples of all volunteers were de-identified. Venous blood withdrawal (5 mL) was done before meals. The acquisition of blood serum was carried out by the standard method [41]. 

In this study, the task was to assess the possibility of using the DLS method and the developed spectrometer to assess changes in the size of structures in blood serum in the presence of donor pathologies [42]. In this regard, at this stage of research, donors were not divided according to the type of pathology and its stage inside one experimental group.

The serum was prepared by sedimentation with centrifugation. The size distribution of proteins in blood serum was determined no later than two days after the blood collection. All samples were prepared immediately before experiments by dissolving one volume of blood serum in 9 volumes of veronal buffer (VBS). The total volume of the studied sample was 1 mL. A study of the dynamics of the immune response was carried out after the reaction initiation by an antigen. In addition, one volume of influenza vaccine (Grippol^®^ Plus) was added to the VBS–serum solution.

## 3. Results

### 3.1. Solutions of Albumin

At the first stage of the experiment, size distributions of albumin molecules in water solution (1 g/L in concentration) were investigated. They were measured using both the developed DLS device (Figure 1) and a commercial ZetasizerNano ZS (Malvern) spectrometer. Data are presented for backscattered light. The results are presented in Figure 2. The *y* axis represents the light scattering intensity *N*(*d*) in relative units, which is proportional to particle concentration. The *x* axis represents size *d* of particles with less than 10% error and a confidence level of 95%. The resolution Δ*d* is a non-linear function of size and is shown by width of columns on the histograms. The size diapason for all presented in this work results was *d*_min_ = 0, *d*_max_ = 500 nm. The size diapasons displayed in the following figures were chosen for better data representation. Polydispersity index (PdI) was defined as the standard deviation of the particle diameter distribution in one population divided by the mean particle diameter in this population. For PdI < 0.1 samples were considered monodisperse. Calculated polydispersity index for ZetasizerNano results (Figure 2b) PdI = 0.1; for developed DLS device results (Figure 2a) PdI = 0.01 for single molecules and PdI = 0.04 for aggregates. The spatial coherence factor for albumin Zetasizer measurements was ~0.88; the baseline value was not provided. For developed DLS device results, spatial coherence factor was equal to ~0.85; baseline value we calculated equal to 0.92.

The average size of HSA molecule is equal to 6 nm. This result was obtained by both used devices and coincides with literature data. In Figure 2a, one can detect the presence of albumin aggregates, which is also consistent with the literature. Although, these aggregates were not detected by ZetasizerNano spectrometer, which indicates its insufficient accuracy when studying mixtures of particles of different sizes.

Effective molecular sizes of albumin at various pH values were studied using a DLS-spectrometer (Figure 1). It is known that proteins aggregation stability depends on the pH of solution. The isoelectric point for HSA is equal to pH = 4.8, at this point albumin is prone to aggregation. In our experiment, the acidity of the solutions varied from strongly acidic (pH = 1.7) to alkaline (pH = 8). pH was measured by digital pH-meter SanXin PHB-3. Figure 3 shows the experimentally obtained dependence of the protein size on the pH of the solution. Figure 4 shows examples of size distribution of albumin proteins and their aggregates at pH = 4.1 ((Figure 4a) and pH = 3.5 (Figure 4b).

It can be seen that the maximum size of albumin aggregates was obtained near the isoelectric point of the protein, which indicated the formation of conglomerates. The obtained result confirms the possibility of studying the dynamics of proteins aggregation using the described method.

The next phase of the study was devoted to demonstrating the ability to detect protein binding to other particles. Figure 5 shows the distribution of nanoparticles and their aggregates in a mixture of gold nanoparticles (d = 20 nm) and albumin. The average size of detected aggregates in the studied mixture was 26 nm, and it reached 38 nm for a smaller amount of aggregates. A small number of structures with size below the average we consider as non-aggregated albumin molecules. We choose neutral pH = 7 for studied solutions.

At this stage of experiments, we demonstrated the possibilities of the proposed DLS method and spectrometer for determining of nanoparticles sizes in mono- and polydisperse solutions using albumin solutions. These results are in accordance with the data received by other researchers with the use of other methods.

### 3.2. Size of Structures in the Blood Serum

At the next stage of the studies, the sizes of molecules and molecular structures in the blood serum of healthy donors and donors with various pathologies were determined. Figure 6 shows the average result of a study of 15 healthy donors. Both individual proteins and their fractions with sizes of about 100 nm could be observed. These sizes are characteristic of medium-sized circulating immune complexes. The sizes of proteins and aggregates are typical for a healthy donor’s blood composition. Aggregates with sizes 31–150 usually stand for low-molecular and high-molecular CICs. The relative concentration of detected CICs is lower compared to the concentration of single proteins, which also indicates the normal immune status of donors.

Figure 7 shows the size of the structures in the blood serum of donors with pathologies. Figure 7a shows the result of measuring the size of structures in the blood serum of donors with non-insulin-dependent diabetes mellitus. According to published data [43], the average concentration of CICs in the blood of patients with diabetes mellitus (insulin-dependent and non-insulin-dependent) is up to 2 times higher than a normal one. In our work, an increase in the relative concentration of CICs is also noted. In addition, an increase in the average size of the CICs was found, which is also an indicator of the pathological process.

Figure 7b shows the average result of measuring the size of structures in the blood serum of donors with cancer. A significant increase in the size and concentration of CICs for all donors can be noted. In addition, a small amount of CICs with sizes below average was observed in the blood serum. As noted in the literature review, it could lead to the development of concomitant disorders of the body. This fact requires further investigation.

This study found that the molecular compositions of healthy donors’ blood serum do not differ significantly from each other, while for donors with diseases, the relative concentration of individual molecular fractions changes dramatically. The increase in the concentration and size of the CICs up to 200 nm was also noted in blood serum of donors with diseases. For presented data sets, Wilcoxon’s rank sum test analysis was performed, and data were considered significant with a *p*-value of less than 0.05 (*p* = 0.03 for diabetes mellitus and *p* = 0.0004 for oncology).

The experimental data obtained in this study correlate with the known data on the size of structures in the blood serum of both healthy people and donors with diseases [10,16]. To establish patterns, further research is needed with a wider representative group of donors. However, the obtained data indicate the applicability of the proposed dynamic scattering spectrometer for assessing human condition by studying the size distribution of structures in blood serum. Moreover, due to the simplicity of research and no need to use specific biochemical markers, this method can become a promising tool in modern medical laboratory diagnostics.

### 3.3. Analysis of the Dynamics of Immune Activation

The study of immune responses was carried out using the blood serum of healthy donors and donors with disorders of the immune response provoked by different diseases. We used an antigen solution to initiate immune reactions in the experiment. All experiments were performed at room temperature, which slowed down the response of the immune system. The dimensions of the structures in the blood serum were observed for 245 s following the addition of the antigen to the blood serum solution. 

Figure 8a presents the averaged results of a study of immune activation in the sera of 10 healthy donors. It can be noted that after adding the antigen, agglomerates with sizes from 40 nm were formed. There is also an increase in the size of complexes up to more than 200 nm immediately after the addition of antigen in the serum. This may be due to the process of their aggregation as a result of immune response activation. There is a gradual increase of structure sizes from 8 to 40 nm in the course of the immune response. Starting from about 175 s after antigen was added to the serum, no dynamic processes were observed, and we can conclude that the process of immune complexes self-assembly on the antigen was finished.

Figure 8b presents the averaged results of a study of immune activation of sera from eight donors with immune response disorders provoked by different diseases. After initiation, the structures with sizes 18–60 nm wer formed. The immune reaction dynamics in this case differ from the reaction of healthy donors due to violations of the immune response.

To observe separately the activation reaction of a specific immune system, one volume of ethylene glycol triamine (EGTA) was added to the blood serum of healthy donors and donors with diseases. EGTA binds calcium ions and blocks the classical pathway of complement system activation. Figure 9 shows the results of the immune response activation in the blood serum of healthy donors (Figure 9a) and donors with impaired immune response (Figure 9b).

It can be noted that when EGTA is added to the blood serum of healthy donors, the general nature of the immune response course does not change in comparison with the reaction of the immune system without EGTA. There are fewer immune complexes with sizes of 40–60 nm; these sizes probably correspond to the complexes of the complement system that are formed upon activation of the classical pathway of the complement system. In addition, with partial blocking of the activation reaction using EGTA (blocking of C1 component), a delayed immune response is observed. We associate it with alternative pathway activation. It is triggered only after covalent binding of component C3b, which is formed as a result of spontaneous hydrolysis. In this case, we suppose that the formation of membrane attack complex (aggregation dynamics) will be partially delayed.

In the blood serum of donors with impaired immune response, with the addition of EGTA and after activation of the immune system using an antigen, no immune reaction is observed. This can be attributed to the complete blocking of the immune response under the influence of EGTA and immune disease.

The experiments were carried out to simulate the processes of self-organization and the ordering of the immune complexes in vivo in the flow of immune reactions. Differences in immune response revealed in samples with active and inactivated complement system as well as disorders of immune response caused by diseases allow us to conclude that it is possible to analyze the processes of ordering various molecular complexes by the DLS method. The studied dynamic activation processes are responsible for the functional properties of the most important proteins.

## 4. Discussion

The results demonstrated in this paper revealed the possibility to detect molecular aggregation at immune system activation by the proposed dynamic light scattering method. The change in the size of detected aggregates upon immune system activation corresponds to the ability of blood serum molecules to form ordered structures and immune complexes. Studies conducted with the proposed method made it possible to observe the molecular complexes formation dynamics in the blood serum, as well as to detect differences in immune reactions in healthy donors and donors with impaired immune systems. Such information may help to determine the functionality of the patient’s immune system and conduct diagnostic tests for certain immune diseases.

Very important results on changes in the size and concentration of structures in the blood serum of the donors with diseases were also demonstrated. The received data are in agreement with what is already known from the literature that shows applicability of the offered spectrometer of dynamic scattering for estimation of a human condition by studying particle size distribution of structures in blood serum without the necessity of the use of special biochemical reagents. It provides a possibility to include this relatively cheap and convenient method in routine biomedical research.

## 5. Conclusions

The DLS spectrometer developed by the authors, the measurement procedure, and the data processing algorithm made it possible to determine the size distribution of protein fractions in blood serum with an error less than 10%. The proposed DLS spectrometer is a small-sized portable device. It can be used even in mobile laboratories, which is especially important in remote regions. In this work, we demonstrated the possibility of a dynamic light scattering method to determine the immune status and to detect pathologies of the human body. Despite the fact that the obtained results require further investigation, the potential of the proposed method for biomedicine is obvious.

## Figures and Tables

**Figure 1 biology-09-00123-f001:**
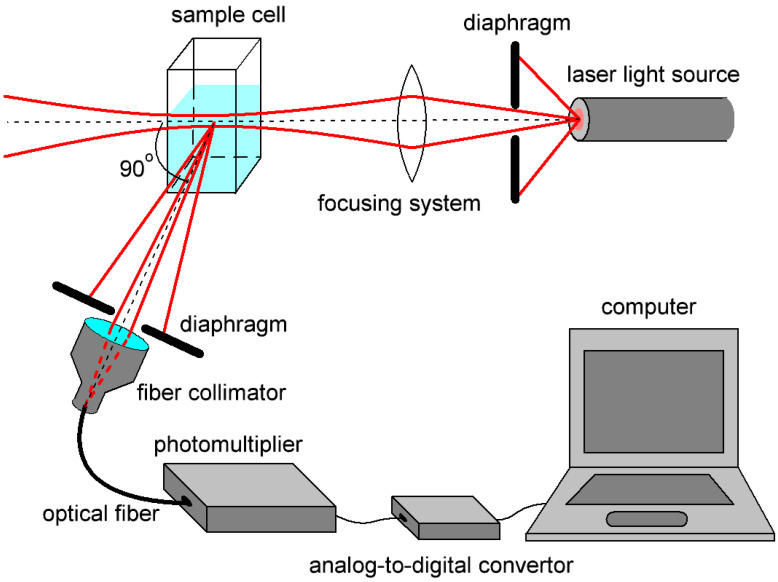
Scheme of a dynamic light scattering spectrometer.

**Figure 2 biology-09-00123-f002:**
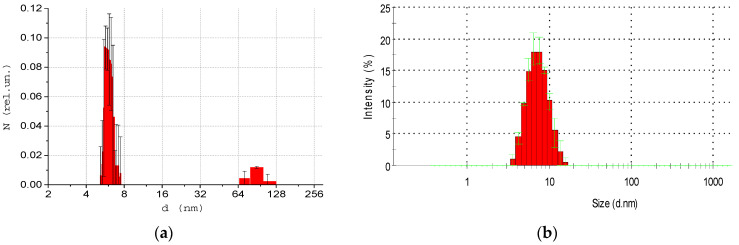
Albumin molecule diameter obtained by the developed DLS device (**a**) and ZetasizerNano spectrometer (**b**). Error bars indicate standard deviation (SD) of the mean. The number of measurements *M* = 10.

**Figure 3 biology-09-00123-f003:**
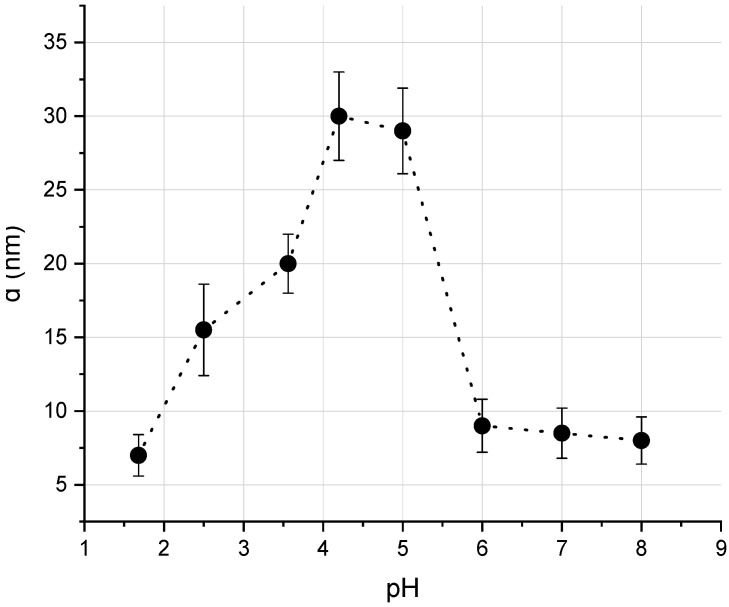
A plot of the diameter of albumin protein aggregates vs. pH of solution. Data are displayed as mean ± SD, *M* = 10.

**Figure 4 biology-09-00123-f004:**
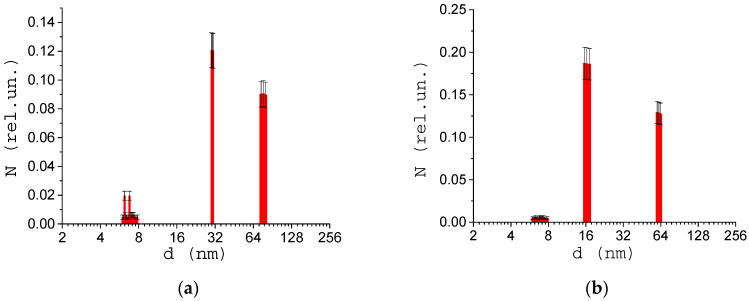
Albumin molecule diameter obtained by the developed dynamic light scattering (DLS) device. pH = 4.1 (**a**), pH = 3.5 (**b**). The number of measurements *M* = 10.

**Figure 5 biology-09-00123-f005:**
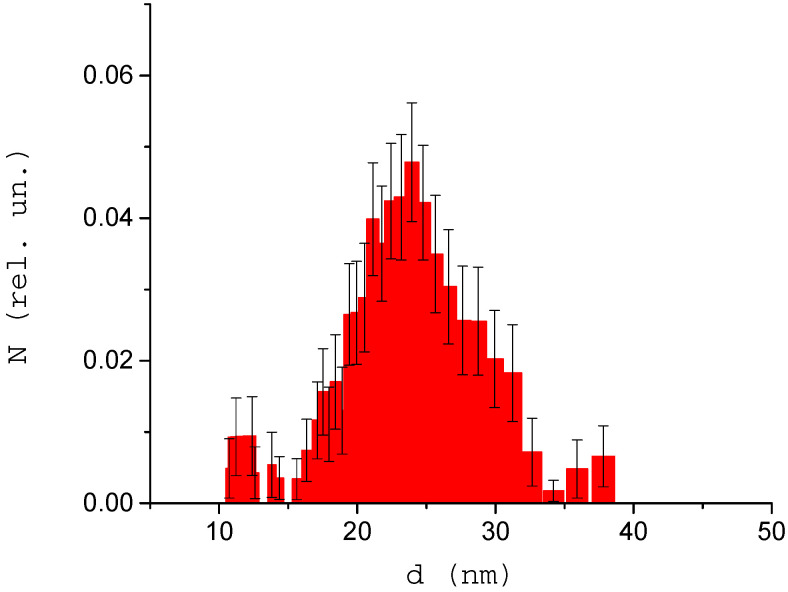
Distribution of size of structures in a mixture of monodisperse gold nanoparticles and albumin protein. The number of measurements *M* = 10. PdI = 0.16.

**Figure 6 biology-09-00123-f006:**
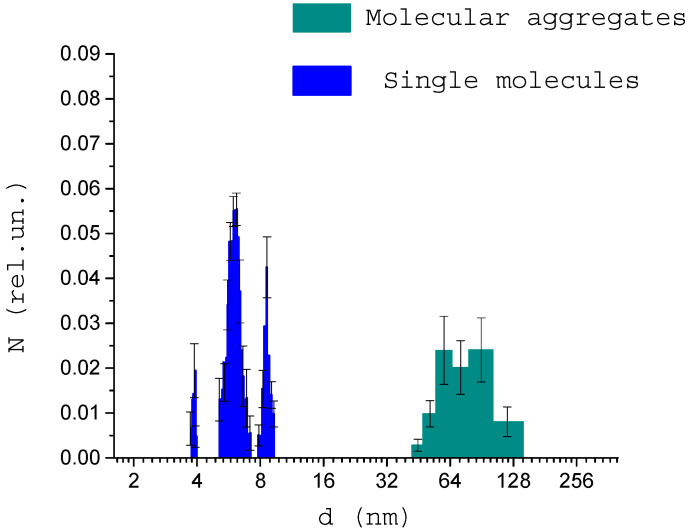
Distribution of size structures in the blood serum of healthy donors.

**Figure 7 biology-09-00123-f007:**
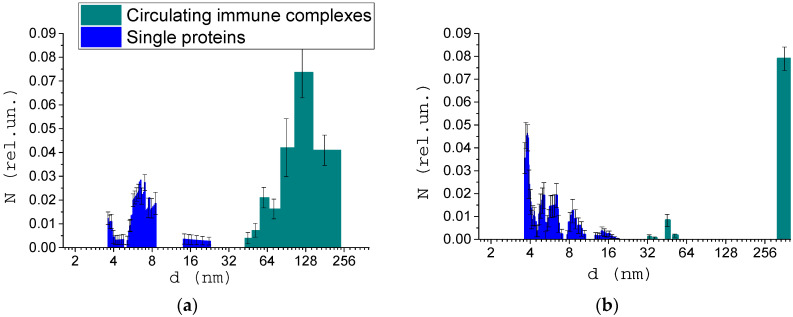
Size distribution of structures in the blood serum of donors with diabetes mellitus (**a**) and donors with cancer (**b**).

**Figure 8 biology-09-00123-f008:**
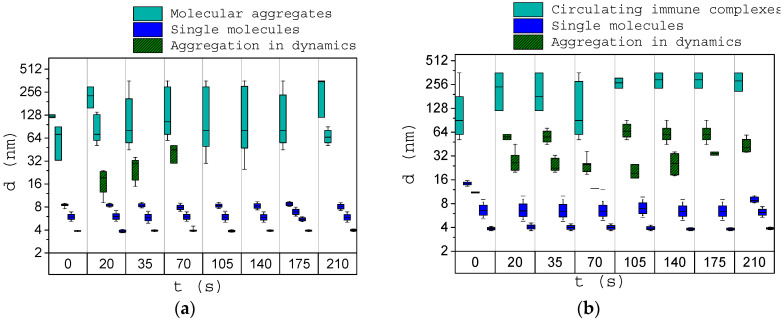
Size of structures in the blood serum before (0 s) and after (20–210 s) antigen addition for healthy donors (**a**) and oncology patients (**b**). Error bars indicate the range of size distributions (minimal and maximal sizes of the certain protein fraction).

**Figure 9 biology-09-00123-f009:**
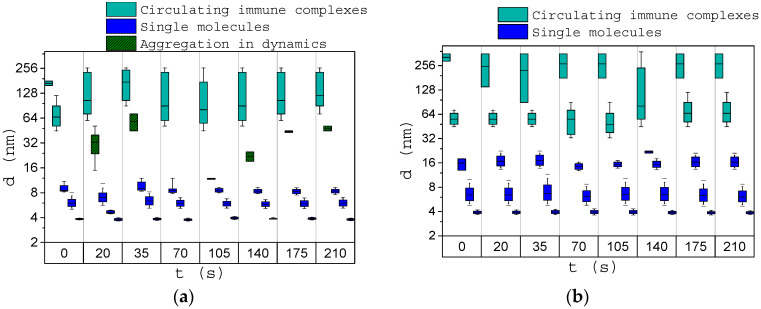
Size of structures in the blood serum with EGTA before (0 s) and after (20–210 s) antigen adding for healthy donors (**a**) and oncology patients (**b**).

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
