# Peer review of "Molecular Aggregation in Immune System Activation Studied by Dynamic Light Scattering"

_biology, 2020, doi:10.3390/biology9060123_

Round 1

Reviewer 1 Report

The manuscript leaves the impression of good work. I've got several questions and suggestions to authors.

Page 2:

1)  Could you please put some explanations on why specific spectral intensity peak corresponds to a particular protein?

Page 3 concerning your setup: 

2) What is the limitation for sample thickness?

3) Usually, it is a bit tricky to use photomultiplier in such a manner as the signal is affected by the several types of noise (shot noise etc.). Configurations with lock-in amplifier implementing synchronous detection are more common. Could you discuss why do you stick to the selected approach, and how does that limit you in your measurements? 

4) What was the coherence length for the laser? Could you please discuss how that parameter affects the quality of your measurements?

5) How does the slit size affect the parameters of the registered spatial scattering coherence?

6) Could you please give type and parameters of the fibre. The type of fibre, whether it is single or multimode, can be necessary for the interpretation of the obtained results. 

7) Did you consider using a NIR laser as the absorption for one would be even lower?

8) Give the path length of the cuvette. 

9) Page 4 in " The proposed scheme of DLS spectrometer allowed us to achieve high SNR ..." Put here some numerical estimation for SNR.

10) You mentioned Tikhonov regularization multiple times. Would you say a couple of words about why do you use such an approach in your data processing? 

11) Page 7 in " The sizes of proteins and aggregates are typical for healthy donor's blood composition..." Why do you think that were CIC but not another protein aggregation from blood serum. What are the specific types of CIC in the scope of your method?

12) Provide specific information on the type and stages of cancer development in the selected group of volunteers. 

Author Response

The manuscript leaves the impression of good work. I've got several questions and suggestions to authors.

Page 2:

Point 1: Could you please put some explanations on why specific spectral intensity peak corresponds to a particular protein?

Response 1: In this sentence, we are talking about the spectral density of the detected light scattering signal. The fluctuation of the density is connected with the Brownian motion of particles, so there will be the Doppler shift in its frequency F, as a result of the particle motion. This shift depends on particle mass and size, and albumins and globulins have different sizes, so different shifts, and frequencies correspond to them. We corrected this sentence (page 2) in the paper.

Point 2: Page 3 concerning your setup: What is the limitation for sample thickness?

Response 2: The sample thickness should not be lower than the detected area of scattering. In our case 6.3 mm.

Point 3: Usually, it is a bit tricky to use photomultiplier in such a manner as the signal is affected by the several types of noise (shot noise etc.). Configurations with lock-in amplifier implementing synchronous detection are more common. Could you discuss why do you stick to the selected approach, and how does that limit you in your measurements?

Response 3: The choice of the photomultiplier (PMT) for the spectrometer is explained by careful consideration of the characteristics of the received signals. Main features: very weak scattered radiation signals (~ -70 dBm), low-frequency nature of the received signals (<100kHz), low background illumination due to the use of optical fibers in the optical scheme. With the indicated parameters, the shot noise of the signal and background illumination and other noise components do not lead to a strong decrease in the signal-to-noise ratio. Compared with avalanche photodiode, the PMT used has a significantly higher gain (~ 100 times) and a lower noise factor (~ 5 times). Under these conditions, the use of synchronous detection methods does not seem appropriate because it complicates the optoelectronic circuit and increases its cost. This issue is discussed in detail in our recent article (Ref. 36). We believe that the selected version of the PMT scheme does not limit the possibilities of our measurements and also leads to a gain in signal quality.

Point 4: What was the coherence length for the laser? Could you please discuss how that parameter affects the quality of your measurements?

Response 4: The semiconductor laser (DBR) selected in the work is single-frequency (with Bragg mirrors) with a generation line width of 1 MHz. This ensures a coherence length of more than 100 m, which significantly exceeds the dimensions of the entire spectrometer and the required coherence length, determined by the scattering and photodetection regions. Therefore, this laser does not reduce the quality of the measurements.

Point 5: How does the slit size affect the parameters of the registered spatial scattering coherence?

Response 5: The spatial distribution of the scattered radiation arriving at the photodetector is a speckle picture, which is a result of the interference of many scattered waves with different amplitudes and phases. The nature of the change in intensity in individual small-sized spots of this picture has a random statistical character and is well studied (see Goodman "Statistical Optics"). The size of the speckle pattern recording area is determined by the aperture, which is selected according to the maximum signal-to-noise ratio. The greatest contrast of the signals is achieved when registering one speckle (spot). But at the same time, the signal itself is usually small. An increase in the recording area reduces contrast but increases the signal-to-noise ratio in a certain range. With a large recording area, both contrast and SNR decrease (tending to zero).

Point 6: Could you please give type and parameters of the fibre. The type of fibre, whether it is single or multimode, can be necessary for the interpretation of the obtained results.

Response 6: In this research, we used the step-index multimode fiber optic patch cable with 50 µm core and 400 to 2400 nm wavelength range. Additional fiber collimator with 0.25 numerical aperture was used in pair with the fiber.

Point 7: Did you consider using a NIR laser as the absorption for one would be even lower?

Response 7: Thank you so much for the advice, we might consider it in the future, although using NIR laser complicates the setup, while the effect of absorption in red or infrared wavelength ranges is negligible in presented studies.

Point 8: Give the path length of the cuvette.

Response 8: The optical pathlength of the cuvette was equal to 0.8 cm. We added it to the paper.

Point 9: Page 4 in " The proposed scheme of DLS spectrometer allowed us to achieve high SNR ..." Put here some numerical estimation for SNR.

Response 9: SNR was more than 10 for all detected signals, for example, the SNR value for blood serum samples in minimally used concentration was equal to 28. These estimations were added to the paper.

Point 10: You mentioned Tikhonov regularization multiple times. Would you say a couple of words about why do you use such an approach in your data processing?

Response 10: Thank you for the great suggestion! We added some text to the paper with brief explanation about Tikhonov regularization.

The regularization algorithms do not require the knowledge of the distribution shape and include reasonable physical requirement of non-negativity of results. Among the variety of regularization methods, the CONTIN method was most popular until recent time. Unfortunately, it is very demanding to the choice of regularization parameter and does not allow the resolution of narrow peaks. In this connection, we decided to use a more advanced algorithm based on the Tikhonov regularization method. This algorithm was designed by authors specifically for the study of molecules and molecular aggregates in biological fluids which usually include narrow close standing peaks. Knowing the diapason of molecular aggregate sizes in blood serum we reduced the calculation interval, that allowed us to get higher resolution and reduce time of experiment. In addition, we reduced the number of points in each peak to avoid their blurring. As a result, the developed algorithm allowed analyzing sizes and concentrations of more than 4 components in fluid.

Point 11: Page 7 in " The sizes of proteins and aggregates are typical for healthy donor's blood composition..." Why do you think that were CIC but not another protein aggregation from blood serum. What are the specific types of CIC in the scope of your method?

Response 11: We compared the characteristic size distributions of proteins and their aggregates with the results of other studies. It was reported that the size distribution of healthy donors, aggregates with sizes 31–150 usually stands for low-molecular and high molecular CIC.

Point 12: Provide specific information on the type and stages of cancer development in the selected group of volunteers.

Response 12: In the research, we studied blood serum of volunteers from the staff of the Peter the Great St.Petersburg Polytechnic University with 2nd and 3rd stages of prostate cancer.

Reviewer 2 Report

The paper reports an experimental study on circulating immune complexes to assess the possibility of using DLS to measure particle size and concentration of polydisperse molecular solutions. DLS is a very powerful tool for studying the diffusion behaviour of macromolecules in solution and the present work would be interesting for a broad community interested in Dynamic Light Scattering applications.

The paper is well written but literature on DLS should be more properly cited. There are a number of excellent reviews detailing the theory and applications of DLS that should be cited in the manuscript. See for example papers of Berne and Pecora (1976), Zakharov and Sheffold (2009), Provencher (1982 and 1996), Pusey (1972).

Before recommending publication there are few aspects that need to be clarified by the authors:

  1. The authors assert that a modified DLS method is proposed, but it is not clear how their DLS setup can be considered as innovative. It is widely used by the light scattering community and the higher performance with respect to commercial DLS setup has been previously demonstrated. I believe that the main result of the paper is the experimental proof of reliability of DLS measurements on biological systems. The authors should stress this point and eventually more deeply explain novelties or improvements in their setup.
  2. The authors should explain how they measure and control temperature. Knowledge of accurate temperature is essential for DLS measurements, since the solvent viscosity depends on it, temperature must be kept constant and solvent viscosity must be known for a reliable DLS experiment.
  3. How the authors overcome the heterogeneity of samples? It is crucial for comparison between different preparations and multiple runs. 
  4. Could the authors comment on the polydispersity of their particles?
  5. The authors should show the intermediate scattering functions and provide details about baselines and coherence factors. Did the authors analyse the intermediate scattering functions by ZetasizerNano spectrometer through their regularization method? It could be a more reliable test.
  6. Comparison with CONTIN algorithm results should be better discussed. Number distribution may be ok with good data, but for noisy data or in presence of high molecular weight aggregates the results may be misleading. If the analysis is aimed to study the presence of trace amount of aggregates the intensity distribution is more suitable.
  7. It is quite surprising that ZetasizerNano spectrometer does not detect albumin aggregates. The authors should provide more details on the ZetasizerNano spectrometer detecting system, the given explanation of its accuracy is reductive. Accuracy of different DLS setups should be tested by comparing data collected at the same detection angle. The authors must include a brief discussion on this point.
  8. How the authors calculate the average size of each population? This point should be better discuss.
  9. The authors should show some examples of histograms at different pH. Moreover intermediate scattering functions at different pH could provide interesting details about aggregation.
  10. Figure 4: a small amount of structures with size below the average is found. How the author can explain this peak? In light of results about pH-dependence of protein aggregation, the pH value should be indicated.
  11. It is not clear how the authors explain the aggregation dynamics for healty donors starting with 70 seconds after antigen addition.

Below more detailed comments:

  • p2, line 18: “A growing number of low molecular weight compounds (1-10 nm) and reduction in the number of large proteins (11-30 nm).” If the numbers refer to sizes it should be indicated.
  • p4, line 27: The construction of the sentence explaining Eq. 2 should be revisited to avoid any misunderstanding between the diffusion coefficient and the size of nanoparticles.
  • p4, line 28: an apostrophe in d’ is missing.
  • A larger font size for plot labels should be used.

Author Response

The paper reports an experimental study on circulating immune complexes to assess the possibility of using DLS to measure particle size and concentration of polydisperse molecular solutions. DLS is a very powerful tool for studying the diffusion behaviour of macromolecules in solution and the present work would be interesting for a broad community interested in Dynamic Light Scattering applications.

Point 1: The paper is well written but literature on DLS should be more properly cited. There are a number of excellent reviews detailing the theory and applications of DLS that should be cited in the manuscript. See for example papers of Berne and Pecora (1976), Zakharov and Sheffold (2009), Provencher (1982 and 1996), Pusey (1972).

Response 1: Thank you for the suggestions! We added more citations to the paper. The authors are familiar with these papers and agree that they should be cited in the work.

Before recommending publication there are few aspects that need to be clarified by the authors:

Point 2: The authors assert that a modified DLS method is proposed, but it is not clear how their DLS setup can be considered as innovative. It is widely used by the light scattering community and the higher performance with respect to commercial DLS setup has been previously demonstrated. I believe that the main result of the paper is the experimental proof of reliability of DLS measurements on biological systems. The authors should stress this point and eventually more deeply explain novelties or improvements in their setup.

Response 2: The authors agree with the reviewer that the main result in the paper is the experimental proof of the reliability of DLS measurements on biological systems. The DLS setup was modified to reduce its size by using optical fiber with a special collimator and providing optimal distances and parameters of the device elements, but in general, all main elements are the same as for other similar devices, reported by other researchers. The main difference could be found in the processing algorithm, which the authors described more properly. This algorithm was designed by authors specifically for the study of molecules and molecular aggregates in biological fluids which usually include narrow close standing peaks. Knowing the diapason of molecular aggregate sizes in blood serum we reduced the calculation interval, which allowed us to get higher resolution and reduce the time of the experiment. In addition, we reduced the number of points in each peak to avoid their blurring. As a result, the developed algorithm allowed analyzing sizes and concentrations of more than 4 components in fluid.

Point 3: The authors should explain how they measure and control temperature. Knowledge of accurate temperature is essential for DLS measurements, since the solvent viscosity depends on it, temperature must be kept constant and solvent viscosity must be known for a reliable DLS experiment.

Response 3: The temperature while measurements was set by heat stabilizer with ±0.1°C deviation. All samples were placed in a square-cut glass cell and set in a heat stabilizer before and during measurements.

Point 4: How the authors overcome the heterogeneity of samples? It is crucial for comparison between different preparations and multiple runs.

Response 4: Thank you for the question. In literature, we can find two interpretations of heterogeneity. If we are talking about heterogeneity from the point of compositional heterogeneity of polymers (eg, monomer. dimer), then in our case we consider it as size polydispersity. If we are talking about volume heterogeneity, then we consider the solution homogeneous because of its proper preparation. For simple size measurements, we used long mixing, in some cases for several hours. For dynamic measurements, we used fast steering and controlled the average intensity of transmitted light. For homogenous solution the average intensity of transmitted light was constant. It indicated the time for measurements start.

Point 5: Could the authors comment on the polydispersity of their particles?

Response 5: Calculated polydispersity index for ZetasizerNano results (Albumin molecules) is PdI= 0.1, for developed DLS device results PdI = 0.01 for single molecules and PdI = 0.04 for aggregates. PdI for gold nanoparticles and albumin protein mixture is equal to 0.16, so this mixture should be considered polydisperse. We added these estimations to the paper.

Point 6: The authors should show the intermediate scattering functions and provide details about baselines and coherence factors. Did the authors analyse the intermediate scattering functions by ZetasizerNano spectrometer through their regularization method? It could be a more reliable test.

Response 6: Intermediate scattering functions were not available from the ZetasizerNano spectrometer, so in this study, we did not analyze the data. We found it possible not to provide intermediate scattering functions because they do not carry additional information. The spatial coherence factor for albumin Zetasizer measurements was ~0.88, the baseline value was not provided by the device. For developed DLS device results spatial coherence factor was equal to ~0.85, the baseline value we calculated equals 0.92. It was added to the paper.

Point 7: Comparison with CONTIN algorithm results should be better discussed. Number distribution may be ok with good data, but for noisy data or in presence of high molecular weight aggregates the results may be misleading. If the analysis is aimed to study the presence of trace amount of aggregates the intensity distribution is more suitable.

Response 7: Thank you for such a useful comment! Indeed, the authors presented the results of the ZetasizerNano measurements as a number distribution. We changed this figure to intensity distribution. All experimental results, provided by our device were calculated as intensity distributions as well.

Point 8: It is quite surprising that ZetasizerNano spectrometer does not detect albumin aggregates. The authors should provide more details on the ZetasizerNano spectrometer detecting system, the given explanation of its accuracy is reductive. Accuracy of different DLS setups should be tested by comparing data collected at the same detection angle. The authors must include a brief discussion on this point.

Response 8: Zetasizer Nano ZS spectrometer uses Non-Invasive Backscatter optics. It is declaimed that these optics has significantly better performance than systems using 90-degree scattering optics. But provided experiments while previous work showed that for protein sizing especially in blood serum is more productive under 90-degree angle. As it was impossible to set 90-degree angle in ZetasizerNano, we presented final results using both spectrometers. We suppose that the main problem in Zetasizer Nano ZS spectrometer accuracy is caused by data processing.

Point 9: How the authors calculate the average size of each population? This point should be better discuss.

Response 9: When calculating particle sizes using the developed algorithm, the maximum number of points in the considered particle population is specified, usually, we used 20 points. By successive iterative solution of the problem, the number of points is reduced and the central values of particle sizes in the population are found that correspond to the maximum scattering intensity.

Point 10: The authors should show some examples of histograms at different pH. Moreover intermediate scattering functions at different pH could provide interesting details about aggregation.

Response 10: We added examples of size distributions for two pH values, which are most representative. We agree with the reviewer, that it provides more reliable information.

The authors described the processing algorithm in more detail, so decided it possible not to provide intermediate scattering functions, because it is an intermediary stage of algorithm analysis.

Point 11: Figure 4: a small amount of structures with size below the average is found. How the author can explain this peak? In light of results about pH-dependence of protein aggregation, the pH value should be indicated.

Response 11: A small number of structures with size below the average we consider as non-aggregated albumin molecules. We choose neutral pH = 7 for studied solutions. We added this information to the paper.

Point 12: It is not clear how the authors explain the aggregation dynamics for healty donors starting with 70 seconds after antigen addition.

Response 12: In the case of studies of the immune response in healthy donors starting with 70 seconds after antigen addition, aggregation reactions are not observed. When blocking of the classical pathway immune activation (blocking of C1 component) reaction using EGTA, immune response after 70 seconds is observed. We associate it with alternative pathway activation. It is triggered only after the covalent binding of component C3b, which is formed as a result of spontaneous hydrolysis. In this case, we suppose that the formation of the membrane attack complex (aggregation dynamics) will be partially delayed. While the formation of Ig immune complexes is not affected by EGTA, so the activation of the specific immunity starting with 20 seconds is still observed.

Point 13: Below more detailed comments:

p2, line 18: “A growing number of low molecular weight compounds (1-10 nm) and reduction in the number of large proteins (11-30 nm).” If the numbers refer to sizes it should be indicated.

p4, line 27: The construction of the sentence explaining Eq. 2 should be revisited to avoid any misunderstanding between the diffusion coefficient and the size of nanoparticles.

p4, line 28: an apostrophe in d’ is missing.

A larger font size for plot labels should be used.

Response 13: The paper was modified accordingly to the comments.

Round 2

Reviewer 2 Report

The paper has been sufficiently improved and can be accepted in the present form. However, if the authors believe that the intermediate scattering functions do not carry additional information to the paper, they should at least refer to previous papers in which the intermediate scattering functions for their developed DLS device were provided.